# Experimental and analytical pipeline for sub-genomic RNA landscape of coronavirus by Nanopore sequencer

Bo-Jia Chen,[1] Ching-Hung Lin,[2] Hung-Yi Wu,[2] James J. Cai,[3] Day-Yu Chao[1,4,5]

**ABSTRACT** Coronaviruses (CoVs), including severe acute respiratory syndrome coronavirus 2, can infect a variety of mammalian and avian hosts with significant medical and economic consequences. During the life cycle of CoV, a coordinated series of subgenomic RNAs, including canonical subgenomic messenger RNA and non-canonical defective viral genomes (DVGs), are generated with different biological implications. Studies that adopted the Nanopore sequencer (ONT) to investigate the landscape and dynamics of viral RNA subgenomic transcriptomes applied arbitrary bioinformatics parameters without justification or experimental validation. The current study used bovine coronavirus (BCoV), which can be performed under biosafety level 2 for library construction and experimental validation using traditional colony polymerase chain reaction and Sanger sequencing. Four different ONT protocols, including RNA direct and cDNA direct sequencing with or without exonuclease treatment, were used to generate RNA transcriptomic libraries from BCoV-infected cell lysates. Through rigorously examining the k-mer, gap size, segment size, and bin size, the optimal cutoffs for the bioinformatic pipeline were determined to remove the sequence noise while keeping the informative DVG reads. The sensitivity and specificity of identifying DVG reads using the proposed pipeline can reach 82.6% and 99.6% under the k-mer size cutoff of 15. Exonuclease treatment reduced the abundance of RNA transcripts; however, it was not necessary for future library preparation. Additional recovery of clipped BCoV nucleotide sequences with experimental validation expands the landscape of the CoV discontinuous RNA transcriptome, whose biological function requires future investigation. The results of this study provide the benchmarks for library construction and bioinformatic parameters for studying the discontinuous CoV RNA transcriptome.

**IMPORTANCE** Functional defective viral genomic RNA, containing all the *cis*-acting elements required for translation or replication, may play different roles in triggering cell innate immune signaling, interfering with the canonical subgenomic messenger RNA transcription/translation or assisting in establishing persistence infection. This study does not only provide benchmarks for library construction and bioinformatic parameters for studying the discontinuous coronavirus RNA transcriptome but also reveals the complexity of the bovine coronavirus transcriptome, whose functional assays will be critical in future studies.

**KEYWORDS** bovine, coronavirus, discontinuous RNA, Nanopore sequencer

Coronaviruses (CoVs), belonging to the subfamily *Coronavirinae*, family *Coronaviridae*, order *Nidovirales*, are positive-sense, single-stranded RNA viruses with a genome size of about 26–32 kilobases (kb) (1, 2). Based on the genomic and phylogenetic structures, CoVs can be further divided into four genera, including *Alphacoronavirus*, *Betacoronavirus*, *Gammacoronavirus*, and *Deltacoronavirus* (3). Among the seven coronaviruses (HCoVs) known to infect humans, HCoV-OC43, SARS-CoV-1, MERS-CoV, and

Address correspondence to Day-Yu Chao, dychao@nchu.edu.tw.

The authors declare no conflict of interest.

See the funding table on p. 18.

*This article is dedicated to Dr. David A. Brian.*

SARS-CoV-2 belong to *Betacoronavirus*, whereas HCoV-229E and HCoV-NL63 belong to *Alphacoronavirus* (4). Due to the biosafety concern of several HCoVs requiring studying under the biosafety level 3 (BSL-3) condition, bovine coronavirus (BCoV), which also belongs to the genus *Betacoronavirus,* has been used as a model under BSL-2 laboratory to explore the impacts of antivirals on the genome structure, pathogenicity, and virus fitness (5).

The structure of the coronavirus genome consists of a 5′ cap, a 5′-untranslated region (UTR), open reading frames (ORFs), and a 3′-UTR including a poly(A) tail. Upon cell entry, the full-length genomic RNA is responsible for the translation to produce nonstructural proteins (nsps) from two ORFs, ORF1a and ORF1b, as well as for replication and transcription, which are mediated by nsp12 harboring RNA-dependent RNA polymerase activity (6). During the life cycle of CoV, they also generate canonical discontinuous subgenomic RNAs (can_sgmRNA or sgmRNA), which will be further translated into four structural proteins [spike protein (S), envelope protein (E), membrane protein (M), and nucleocapsid protein (N)] and several accessory proteins (7, 8). The discontinuous transcription into sgmRNA is mediated by the transcription-regulatory sequences (TRS) containing the common 5′ "leader" sequence of ~70 nucleotides fused to the "body" sequence from the downstream part of the genome (9, 10). Other than can_sgmRNA, defective viral genome (DVG) with non-canonical truncated viral RNAs was observed, some of which were considered as defective interfering RNA (DiRNA) (11–13). There are different kinds of cis-acting elements required for viral genome transcription, gene expression, and pathogenesis in the 5′ and 3′ termini of the coronaviral genome. The *cis*-acting elements in the 5′ terminal of the genome are composed of multiple stem-loops (SLs), including SL I to VII. While the cis-acting elements in the 3′ end of the genome consist of a bulged stem-loop, pseudoknot, hypervariable region, and poly(A) tail (13, 14). Functional DVGs (so-called DiRNA), if they contain the cis-acting elements required for translation or replication, may play different roles in triggering cell innate immune signaling, interfering with sgmRNA transcription/translation, or assisting in establishing persistence infection (15, 16).

The S protein, which mediates the binding with cellular receptors for infection, is a characteristic feature of the *Coronaviridae* family. Both SARS-CoV-2 and SARS-CoV bind to a common human receptor, angiotensin-converting enzyme 2 (ACE2), which is also the receptor for other human CoVs except MERS-CoV (17). The S protein consists of two subunits: the S1 unit at the N-terminus of the S protein forms the head that contains the receptor-binding domain (RBD) and is responsible for cellular receptor binding, whereas the S2 unit is present in the stalk of the S protein, mediating the fusion process for viral entry (17). These two subunits are separated by the site that contains a furin cleavage motif and is cleaved by the transmembrane serine protease TMPRSS2 in the virus-producing cell (18). This cleavage activates the S2 subunit trimers to fuse viral and host lipid bilayers, releasing the viral ribonucleoprotein complex into the cell. Amino acid variations in human ACE2 proteins have been suggested to mediate RBD binding affinity, which could either enhance or inhibit virus entry (19). As such, vesicles designed to carry the S protein or RBD could be used to antagonize virus entry (20–22). Alternatively, extracellular vesicles, derived from stem cells that carry ACE2, could be used to treat infections by coronaviruses (23, 24).

To better understand the underlying mechanism of DiRNA or DVGs, the landscape of viral RNA transcriptome upon infection can be revealed using high-throughput sequencing approaches. Generally, the gene expression levels are well-captured using short-read sequencing (SRS) like the Illumina sequencer (25, 26), but SRS techniques have difficulties in identifying multi-spliced transcripts or transcriptional start/end site isoforms due to its limited read length and the fact that RNA molecules have to be reverse transcribed before sequencing (27). Nanopore sequencer (ONT), one of the long-read sequencing (LRS) techniques, offers the possibility of sequencing long reads and, most importantly, the native RNA molecules without the requirement for polymerase chain reaction (PCR) amplification, therefore reducing PCR-induced biases

in the estimation of expression levels (28). Several studies have adopted the Nanopore sequencer to identify non-canonical RNA junctions, and unexpected RNA isoforms have been found in coronavirus research (29–33). Although this technique is cost-effective with yields in both higher throughput and longer reads, the published studies have neither investigated how different parameters in bioinformatic approaches affect the spectrum and accuracy in detecting DVGs nor validated the accuracy of identified viral RNA transcriptional isoforms experimentally. It is unclear if the identified DVGs with non-canonical junctions are broadly representative of CoV biology, including BCoV, or if they are a result of library- or bioinformatics-specific artifacts (34).

In this study, we used both ONT MinION direct RNA sequencing (DRS) and direct cDNA sequencing to evaluate BCoV transcriptomes. Since the majority of host messenger RNA (mRNA) or viral RNA transcripts are protected from degradation by m7Gppp cap and triphosphate (35), the 5′ phosphate-dependent exonuclease, by removing the RNA population with 5′ monophosphate group such as ribosomal RNA (rRNA) and transfer RNA (tRNA), was tested in this study for its influence on BCoV transcriptome. Our intentions are to investigate the effect of (i) RNA samples with or without being processed by 5′ phosphate-dependent exonuclease; (ii) native RNA or cDNA library construction; and (iii) bioinformatics procedures, including denoise and classification, on detecting the discontinuous RNA transcriptomic landscape of BCoV, especially DVGs. To validate the results from Nanopore sequencing, we further confirmed by reverse transcription PCR (RT-PCR) and picked single colonies for Sanger sequencing. A previously unreported discontinuous RNA transcript population was identified. An analytical pipeline to reveal the transcriptome landscape is proposed for future study.

## RESULTS

### Overall data characteristics and BCoV read precision comparison

We utilized two ONT library preparation workflows in this study (Fig. 1). What they have in common is that the RNA or cDNA molecules were sequenced directly without PCR amplification. The yields from the different protocols varied with the total reads ranging from approximately 0.14 to 1.68 million sequencing quality-passed reads per library (Table 1). Higher reads from RNA libraries were observed than those from cDNA libraries, possibly due to higher initial RNA input during the library constructions as suggested by the manufacturer's protocols. The read length and read quality were similar within the same library construction strategy (Fig. S1A and B). The ONT reads were aligned to three reference genomes using minimap2 (see Materials and Methods). As the unmethylated RNA calibration standard (RCS), used to assess the false detection rate of the methylation calling of RNA molecules, is only offered in the DRS kit, 53.3% and 26.7% of reads on average from protocols #1 and #2, respectively, were mapped to RCS, with a higher RCS read proportion observed in exonuclease treatment libraries (Table 1). No RCS reads were found in protocols #3 and #4 of the cDNA libraries. The exonuclease treatment digests RNA with 5′ monophosphate, mainly rRNA. Therefore, the relatively high percentage of RCS found in protocol #1 could be due to the removal of highly abundant rRNA, which leaves more pores free for RNA sequencing, especially RCS, although the possibility of a higher percentage of RNA degradation by exonuclease treatment could not be ruled out. After subtracting RCS reads from protocols #1 and #2, species differences were observed among protocols with higher proportions of reads mapped to BCoV and lower proportions of reads mapped to human by RNA protocols [analysis of variance (ANOVA), $P$-values are 0.0101 and 0.0193 for human and BCoV, respectively] (Table 1). In particular, protocol #2 (RNA without exonuclease-treatment protocol) showed the highest proportion of reads mapped to BCoV (Fig. S1C). More BCoV reads were mapped to 5′ and 3′ terminals of the viral genome, which reflected the higher abundance of sgmRNA produced from the coronavirus-specific transcription mechanism (1, 2) (Fig. S1D). We then merged two replicates of total reads from each protocol for the following analysis by focusing on BCoV RNA transcripts to increase the sample size.

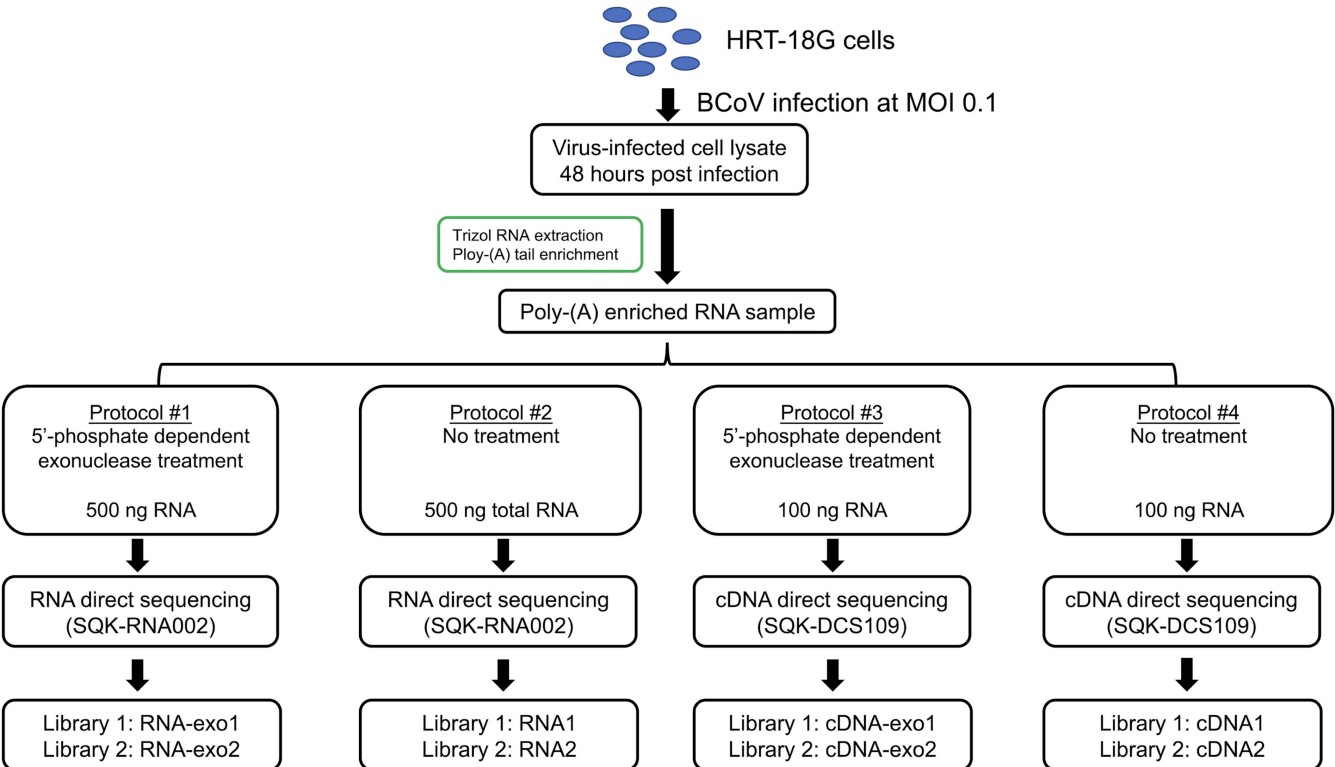

**FIG 1** The experimental design for RNA libraries' constructions by four different protocols to study BCoV RNA transcriptome. Each protocol will have two replicates, named Library 1 and Library 2.

## Discontinuous RNA transcripts' determination

Since our interests were in the landscape of BCoV transcriptomes, we intended to investigate the influence of bioinformatics procedures, including denoise and classification, on detecting various discontinuous RNA transcripts. First, we evaluated different k-mer sizes and their effects on mapped reads of BCoV RNA. Figure S2A shows that only slight influences on the proportion of assigned BCoV reads were observed; however, the number of fragments and their abundancies from each protocol significantly decreased with the increase of the k-mer size cutoff (Fig. 2A). Further investigation on the differences between 8-mer and 15-mer sizes found that the two-fragment reads of discontinuous RNA transcripts identified by 8-mer size were assigned to one-fragment reads by the 15-mer setting (Fig. S2B). Furthermore, when the two-fragment reads were classified as sgmRNA or DVG (as described in the Materials and Methods section), the results showed that the abundance of sgmRNA reads dramatically decreased in RNA protocols but not in cDNA protocols when the cutoff of k-mer size increased (Fig. 2B, upper panel). On the contrary, the abundance of DVG reads showed a significant drop when k-mer size was set at 15 but remained relatively stable even when the k-mer size cutoff continued to increase (Fig. 2B, lower panel). Similar results were also found when investigating the diversity of the recombination types of DVG (Fig. 2C). Further scrutinizing the types and abundancy of DVGs, more than 80% of the types of DVGs found, when k-mer size was set at 8, but "lost" when k-mer size was set at 15, are rare, with the read frequency as low as only one or two reads possessing less than 100 bases in length (Fig. S3A). Their genomic locations were mainly mapped to either the 5′ or 3′ terminals of the BCoV genome (Fig. S3B). Such types of DVGs are not highly reproducible due to their low frequency. These results indicate that the cutoff of k-mer size will significantly alter the landscape of discontinuous RNA transcripts of BCoV generated by Nanopore sequencing, especially using RNA direct sequencing protocol. Based on the results generated here, 8-mer is

**TABLE 1** Sequencing reads of different libraries from different protocols before and after removing RNA calibration standard

| | Mapped reads (with RCS) | | | | | Mapped reads (without RCS) | | | |
|---|---|---|---|---|---|---|---|---|---|
| | Total reads | Human host | BCoV | Yeast | Unmap | Total reads | Human host | BCoV | Unmap |
| **Protocol #1** | | | | | | | | | |
| RNA_exo1 | 1,680,377 | 622,747 | 257,267 | 785,633 | 14,730 | 894,744 | 622,747 | 257,267 | 14,730 |
| | | (37.05%) | (15.31%) | (46.75%) | (0.87%) | | (69.60%) | (28.75%) | (1.65%) |
| RNA_exo2 | 470,534 | 78,370 | 24,355 | 361,007 | 6,802 | 109,527 | 78,370 | 24,355 | 6,802 |
| | | (16.65%) | (5.17%) | (76.72%) | (1.44%) | | (71.55%) | (22.23%) | (6.21%) |
| Subtotal (%) | 2,150,911 | 701,117 | 281,622 | 1,146,640 | 21,532 | 1,004,271 | 701,117 | 281,622 | 21,532 |
| | | (32.59%) | (13.09%) | (53.3%) | (1%) | | (69.81%)[b] | (28.04%)[a] | (2.14%) |
| **Protocol #2** | | | | | | | | | |
| RNA1 | 1,109,516 | 472,678 | 317,746 | 296,525 | 22,567 | 812,991 | 472,678 | 317,746 | 22,567 |
| | | (42.60%) | (28.63%) | (26.72%) | (2.03%) | | (58.14%) | (39.08%) | (2.76%) |
| RNA2 | 242,220 | 97,950 | 73,999 | 63,744 | 6,527 | 178,476 | 97,950 | 73,999 | 6,527 |
| | | (40.43%) | (30.55%) | (26.31%) | (2.69%) | | (54.88%) | (41.46%) | (3.65%) |
| Subtotal (%) | 1,351,736 | 570,628 | 391,745 | 360,269 | 29,094 | 991,467 | 570,628 | 391,745 | 29,094 |
| | | (42.21%) | (28.98%) | (26.65%) | (2.15%) | | (57.55 %)[b] | (39.51%)[a] | (2.93%) |
| **Protocol #3** | | | | | | | | | |
| cDNA_exo1 | 208,409 | 165,055 | 42,138 | 0 | 1,216 | 208,409 | 165,055 | 42,138 | 1,216 |
| | | (79.19%) | (20.21%) | | (0.5%) | | (79.19%) | (20.22%) | (0.58%) |
| cDNA_exo2 | 494,479 | 366,797 | 119,676 | 0 | 8,006 | 494,479 | 366,797 | 119,676 | 8,006 |
| | | (74.17%) | (24.20%) | | (1.6%) | | (74.17%) | (24.20%) | (1.62%) |
| Subtotal (%) | 702,888 | 531,852 | 161,814 | 0 (0.00%) | 9,222 | 702,888 | 531,852 | 161,814 | 9,222 |
| | | (75.67%) | (23.02%) | | (1.31%) | | (75.66 %)[b] | (23.02%)[a] | (1.31%) |
| **Protocol #4** | | | | | | | | | |
| cDNA1 | 827,842 | 594,862 | 226,904 | 0 | 6,076 | 827,842 | 594,862 | 226,904 | 6,076 |
| | | (71.85%) | (27.40%) | | (0.7%) | | (71.85%) | (27.41%) | (0.73%) |
| cDNA2 | 142,109 | 94,393 | 45,516 | 0 | 2,200 | 142,109 | 94,393 | 45,516 | 2,200 |
| | | (66.42%) | (32.02%) | | (1.5%) | | (66.42%) | (32.03%) | (1.55%) |
| Subtotal (%) | 969,951 | 689,255 | 272,420 | 0 (0.00%) | 8,276 | 969,951 | 689,255 | 272,420 | 8,276 |
| | | (71.06%) | (28.09%) | | (0.85%) | | (71.06 %)[b] | (28.09%)[a] | (0.85%) |

[a]ANOVA and Tukey's honest significance (HSD) test (RNA vs cDNA_exo: $P = 0.0177$ and RNA vs RNA_exo: $P = 0.035$).
[b]ANOVA and Tukey's HSD test (RNA vs cDNA: $P = 0.042$, RNA vs cDNA_exo: $P = 0.008$, and RNA vs RNA_exo: $P = 0.029$).

the better setup as the cutoff size for sgmRNA determination without losing abundancy and diversity; however, for DVG determination, the k-mer setup at 15 is optimal for investigating DVGs' abundancy and recombination types to avoid the noise from small fragments.

## Recombination site determination of discontinuous RNA transcripts

Previous studies chose arbitrary gap and fragment sizes while investigating the landscape of discontinuous RNA transcripts of coronavirus (29–31). We focused on the two-fragment reads (i.e., single-junction recombinant DVGs) since they comprised the highest proportion of BCoV discontinuous RNA transcripts (Fig. 2A). Figure 3A and Fig. S4A show the heat maps of the recombination sites of the two-fragment reads. The heat maps among four different protocols looked quite similar with the highest abundant recombination sites near 0 kb of the 5′ end (mainly the leader sequence with ~70 nucleotides), which joined to the 3′ end of the viral genome, representing the can_sgmRNAs. Moreover, the non-canonical discontinuous RNA transcripts (or DVGs) could also be found abundantly within the recombinant sites between ORF-1ab and ORF-N or ORF-M. We then investigated how the different cutoffs in the bioinformatic parameters would affect the abundance of discontinuous RNA transcripts among different protocols. The ridge plots showing the distributions of gap and fragment length are provided in Fig. S4B and C. When the cutoff of the gap size increased, the abundance of discontinuous RNA transcripts dropped, particularly in the RNA protocols (Fig. 3B). A

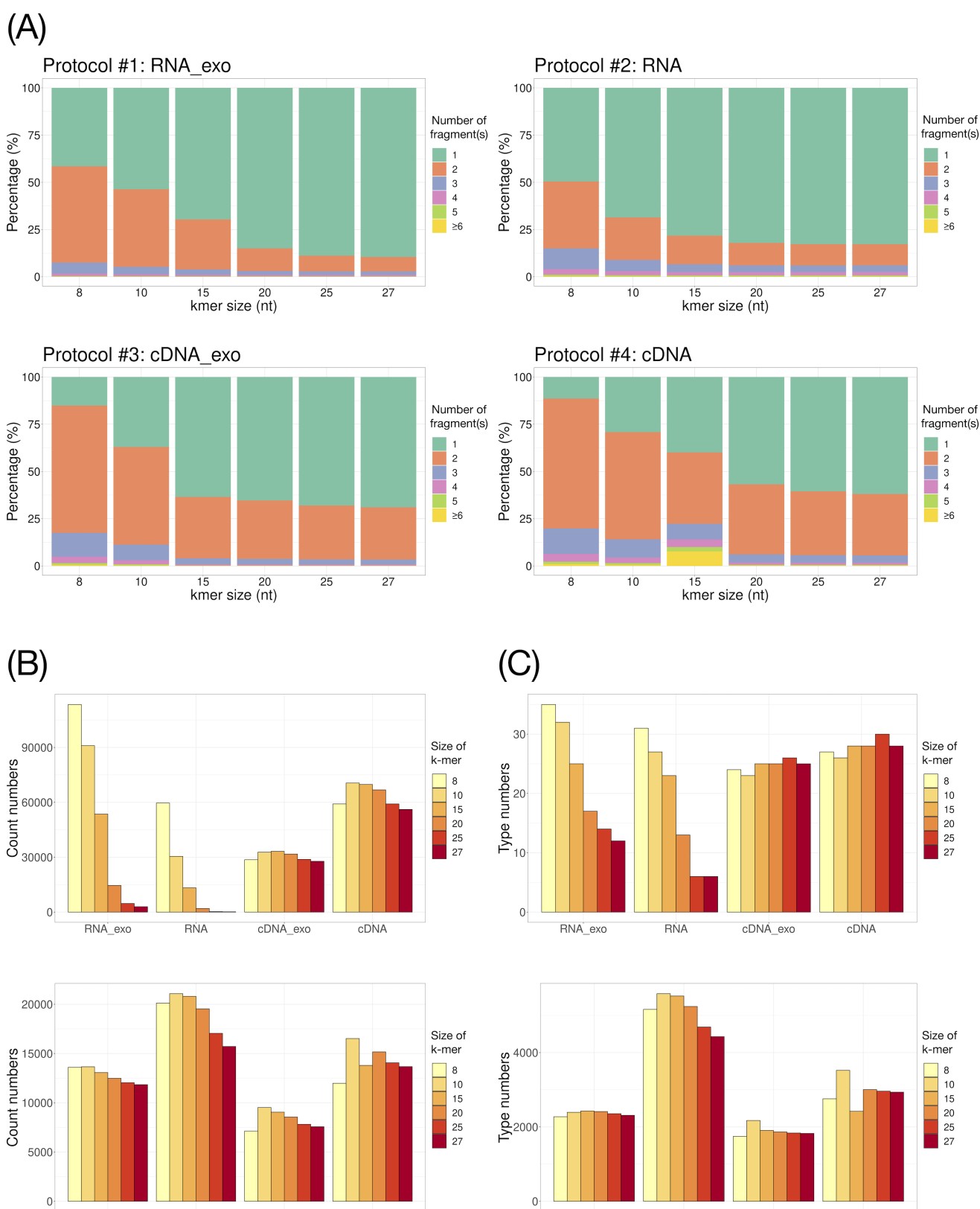

**FIG 2** Comparison of the effects of different k-mer size cutoffs on the discontinuous RNA fragments (A), counts of reads (B), and recombination types (C) mapped to sgmRNA (upper panel in B and C) and DVGs (lower panel in B and C) among different protocols. The percentage is calculated by counts of discontinuous RNA reads divided by total aligned BCoV reads. The RNA fragments of more than six are collapsed into the group of $\geq$6.

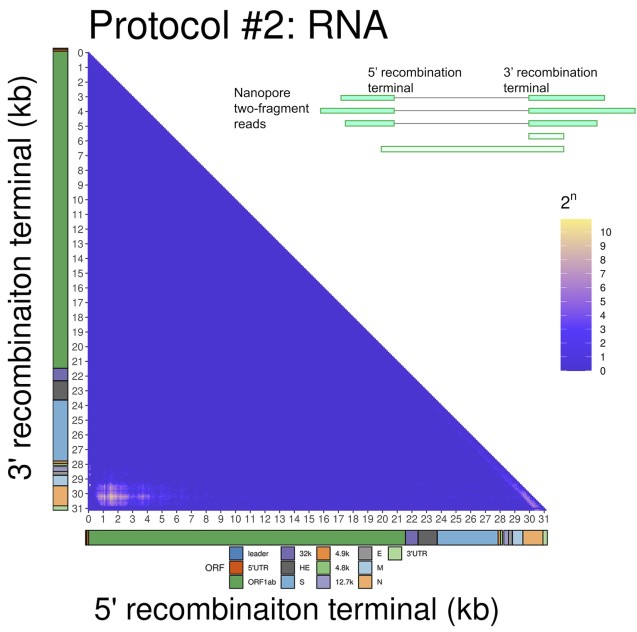

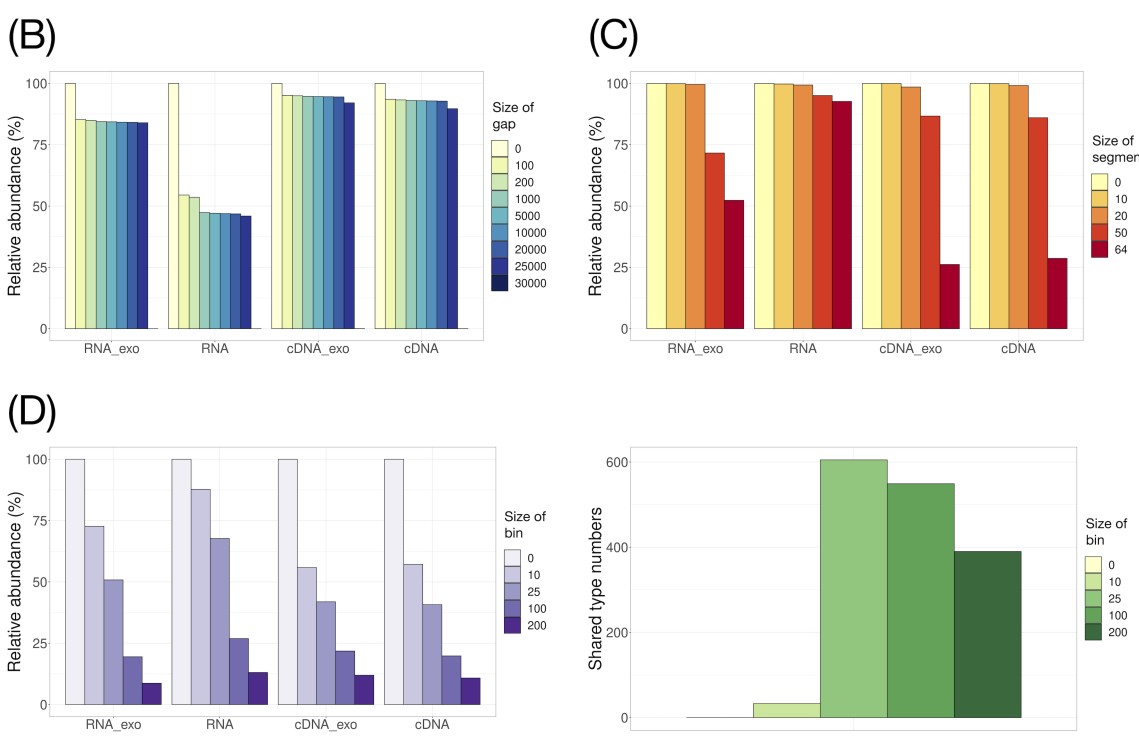

**FIG 3** (A) The heat map of the recombination sites of two-fragment reads based on Protocol #2, RNA library with exonuclease treatment protocol. The upper-right scheme illustrates the representative junction terminal of discontinuous viral RNA transcripts. To better visualize, the coordinate of the bovine coronavirus genome was binned by 100 base pairs, and the x and y-axes were plotted by kilobase. (B–D) Comparison of the effects of different bioinformatic parameters, including gap size (B), segment size (C), and bin size (left panel of D), on the counts of reads and the recombination types mapped to two-fragment discontinuous RNA transcripts of BCoV. Data are transformed into the relative abundance as proportions (%) by calculating the filtered-read counts by gap size and segment length as indicated, divided by the total BCoV read counts. (Right panel of C) Shared recombination DVG types among four different protocols differed by bin size cutoffs.

significant drop in BCoV-mapped reads was observed when segment size increased to 50 bp or higher (Fig. 3C). Therefore, the optimal cutoff for gap size and fragment length will be 200 and 20 bp, respectively, to remove the sequence noise and retain the majority of the informative reads.

Due to the error-prone nature of the ONT sequencer especially at the terminals (36), various transcriptional end sites were commonly generated after BED file transformation. Previous studies used arbitrary bin sizes to determine different types of DVGs (29–31). To decide the optimal bin size, the same types of DVGs in this study were first grouped together based on the same coordinates of the junction sites, which were composed of four binned coordinates of the DVGs. To evaluate the discontinuous RNA transcripts' landscape, we chose different bin sizes to investigate their effects on the types of DVGs and reproducibility among different libraries generated from four different protocols. As expected, the larger the bin size, the fewer types of DVGs were observed among all the protocols (Fig. 3D, left panel). Since all four protocols used the same BCoV strain under the same infection condition, the higher the shared DVGs among the protocols would represent the more desirable bioinformatic condition. As shown in Fig. 3D (right panel), the shared recombination types among the four protocols were the highest when the cutoff of the bin size was set at 100 and gradually decreased as the bin size increased. Therefore, the DVGs were recommended to collapse every 100 base pairs to avoid various recombination sites generated from sequencing errors introduced by Nanopore sequencers while maintaining enough DVG diversity.

To validate if the above bioinformatic parameters of DVGs could be applied to other CoVs, we used the developed analytical pipeline to the publicly available ONT-generated data set of SARS-CoV-2 (https://doi.org/10.1016/j.cell.2020.04.011) (30). Similar patterns were also observed. First, a significant reduction in two-fragment reads was observed with the increasing k-mer size cutoff, although the mapped SARS-CoV-2 reads remained similar (Fig. S5A). While classifying two-fragment reads into sgmRNA or DVGs, the increase in k-mer size resulted in decreasing sgmRNA read counts significantly (Fig. S5B, upper panel). On the contrary, DVG read counts increased when k-mer size increased from 8 to 15 and remained stable afterward (Fig. S5B, lower panel). Similar findings can be observed for the number of recombination types (Fig. S5C). Second, the read counts significantly dropped when the threshold setting of the segment size was larger than 20 (Fig. S5D). Third, the read counts also significantly dropped when the gap size increased from 0 to 100 and remained stable when the gap threshold setting was between 1,000 and 25,000 (Fig. S5E). In summary, our results demonstrated that different parameters in bioinformatic approaches would affect the spectrum and abundance of the identified DVGs in BCoV and SARS-CoV-2. Based on the above results, we summarized and proposed the bioinformatic analytical pipeline for investigating discontinuous RNA transcripts of CoV in Fig. 4.

To further evaluate the sensitivity and specificity of our analytical pipeline, 100 simulated libraries of DVGs with 20,000 reads of each library were generated (Fig. S6A and B), and the results demonstrated that the highest counts of DVG reads can be found when the k-mer size cutoff was set at 15 (Fig. S6C). Through simulation, the highest sensitivity and specificity were reached when the k-mer size was set at 15 with 82.6% and 99.6%, respectively (Fig. S6D).

## Validation of junction sites diversity of single-junction DVGs

As for ONT library construction and sequencing, the intrinsic biases for each read were anticipated to be coming from the poly-A selection, the reverse transcription, and sequencing preferences from the 5′ or the 3′ end of the cDNA (28). To validate the DVGs identified in this study, four types of DVGs were manually chosen based on the following criteria: (i) the recombination site should be at least 1,000 nucleotides away from the recombination sites of sgmRNA at one of the terminals and (ii) are ranked in the top 10 abundances based on the mean proportion calculated from four different protocols. The primers flanking each type of DVG were designed (Fig. 5A). The same

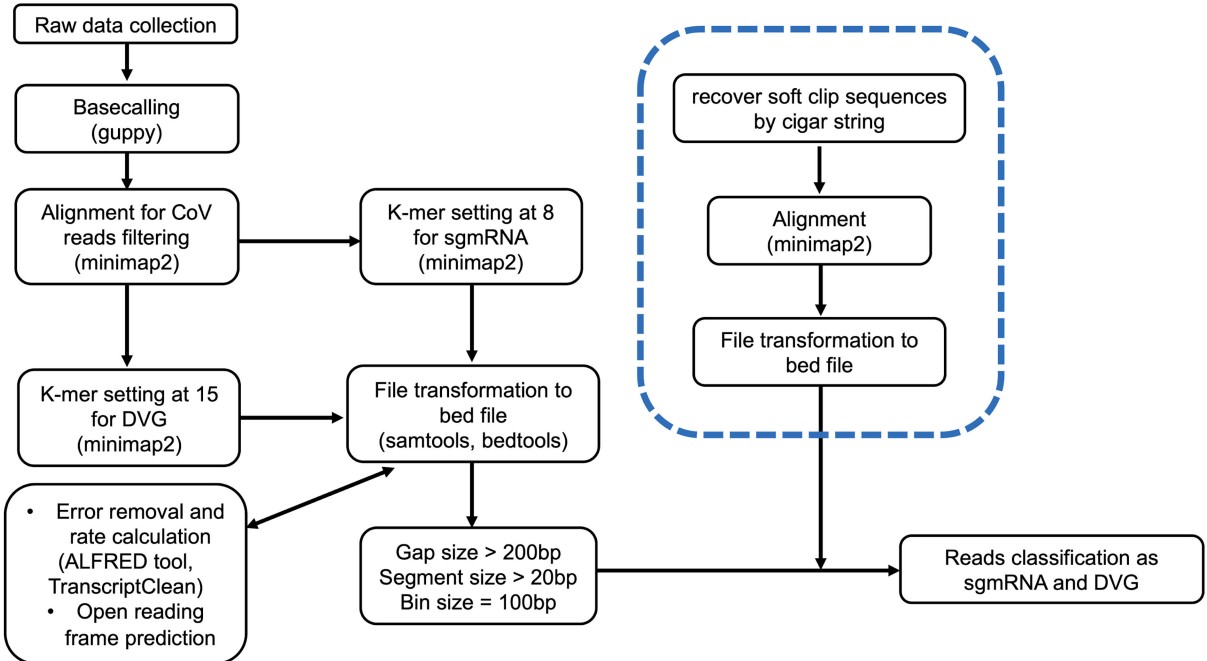

**FIG 4** Proposed bioinformatic pipeline for investigating discontinuous RNA transcriptome of bovine coronavirus, mainly subgenomic messenger RNA and defective viral genomes. The dashed frame indicates the additional step to identify reverse DVGs.

viral-infected cell lysates used to generate the libraries were subjected to RT-PCR and nested PCR. The PCR products of 395–650 bp in length corresponding to each type of DVG were subjected to TA cloning (Fig. 5B). Ten colonies in total with an average of 2.5 colonies from each DVG type were picked and sequenced by traditional Sanger sequencing (Fig. 5C). The sequencing results were first aligned to BCoV viral genome by National Center for Biotechnology Information (NCBI) blast tool, then manually inspected for their recombination site coordinates, which identified three DVGs, two N-sgmRNA, and five reverse DVGs (Table 2).

Among three types of DVGs, clone #1_5, with the two-fragment junction region at 1,700–29,500 of BCoV recombinant coordinate, is translatable and could be found across all four protocols, and the highest abundance was observed from the RNA protocols (Fig. 5D, left panel). The treatment of exonuclease reduced the abundance for both RNA and cDNA protocols, which is consistent with our previous observation (Table 1). On the contrary, clones #3_8 and #3_20 with the junction region at 11,100–29,200 and 11,100–29,500, respectively, can only be found from the four libraries under the k-mer size equal to eight without any filtering but not the k-mer size equal to 15 with filtering. The reason that clones #3_8 and #3_20 could not be detected was due to the short length in the first fragment of the Nanopore reads, which will be clipped by minimap2 under k-mer size equal to 15, and the remaining sequence will be classified as one transcript without recombination. Since both clones #3_8 and #3_20 are not translatable, they could be non-functional DVGs.

Meanwhile, our study also discovered five DVG clone types containing three discontinuous RNA transcripts, whose RNA fragments contained the virus sequence located in the inter-genome and the coordinates of the first fragment were larger than those of the second fragment (Fig. S7A). Such DVG types were previously unidentified and were, herein, named reverse DVGs (Table 2). Only one of them, clone #1_12, can be translated, which could be functional DVGs, although non-translatable DVGs may also have functional interference, such as interfering with sgmRNA translation. Interestingly, such DVGs could not be detected by the current analytical pipeline in Fig. 4. To find out why our current analytical pipeline could not detect reverse DVGs, we further

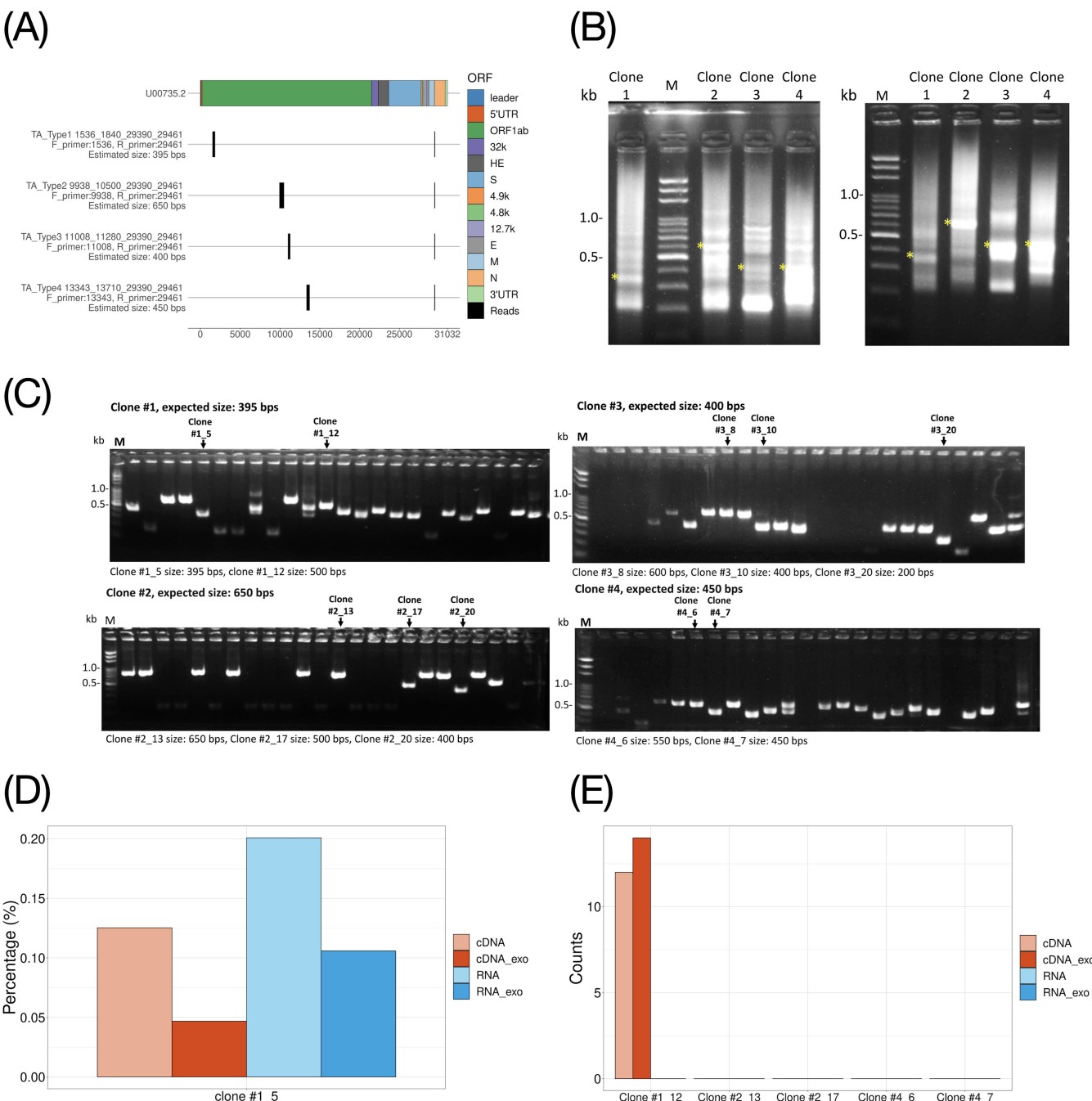

**FIG. 5** Experimental validation of the DVGs from RNA transcriptome of bovine coronavirus-infected cell lysates identified by Nanopore sequencing. (A) Schematic diagram of primer design to amplify four distinguished DVG types shared among four protocols. The transcriptional starting and ending sites of each fragment of four DVG types are depicted as the coordinate vector as (forward primer_5′junction site_3′junction site_reverse primer) based on the full genome of BCoV strain (GenBank accession no. U00735.2). The primer pairs used for PCR amplification and the estimated size are also shown for each DVG type. (B) 1% agarose gel electrophoresis of first (left) and second (right) round PCR. Asterisks indicate the estimated size of PCR products for each DVG type, which were subjected to gel purification. (C) 1% agarose gel electrophoresis analysis of 24 picked clones from each DVG type. The gel-purified DNA products proceeded to TA cloning. Twenty-four colonies from each DVG type were picked for colony PCR. The arrow indicates the clones with the expected size, and clone IDs selected for sequencing were labeled according to Table 2. (D) The relative abundance of the DVG clone #1_5 among four different protocols. Data are transformed into percentages by calculating read counts of such clone type divided by the total BCoV mapped read counts. (E) The total counts of the reverse DVG clones #1_12, #2_13, #2_17, #4_6, and #4_7 identified from TA cloning and Sanger sequencing experiment found in four different protocols.

**TABLE 2** TA cloning results of discontinuous RNA transcripts from bovine coronavirus-infected cells

| Type | Clone | Start1 | End1 | Start2 | End2 | Start3 | End3 | Fragments | Coordinates at bin size 100 | Class | Translatability (start codon) |
|---|---|---|---|---|---|---|---|---|---|---|---|
| 1 | 1_5 | 1,536 | 1,690 | 29,475 | 29,496 | | | 2 | 1,600_1,700_29,500_29,600 | DVG | Y |
| 1 | 1_12 | 1,536 | 1,734 | 6 | 74 | 29,394 | 29,496 | 3 | 1,600_1,800_100_100_29,400_29,500 | Reverse DVG | Y |
| 2 | 2_13 | 9,938 | 10,392 | 10 | 74 | 29,394 | 29,496 | 3 | 10,000_10,400_100_100_29,400_29,500 | Reverse DVG | N |
| 2 | 2_17 | 9,938 | 10,073 | 8 | 74 | 29,394 | 29,496 | 3 | 10,000_10,100_100_100_29,400_29,500 | Reverse DVG | N |
| 2 | 2_20 | 6 | 74 | 29,394 | 29,496 | | | 2 | 100_100_29,400_29,500 | N_sgmRNA | Y |
| 3 | 3_8 | 11,008 | 11,054 | 29,136 | 29,496 | | | 2 | 11,100_11,100_29,200_29,500 | DVG | N |
| 3 | 3_10 | 6 | 74 | 29,394 | 29,496 | | | 2 | 100_100_29,400_29,500 | N_sgmRNA | Y |
| 3 | 3_20 | 11,008 | 11,052 | 29,458 | 29,496 | | | 2 | 11,100_11,100_29,500_29,500 | DVG | N |
| 4 | 4_6 | 13,343 | 13,529 | 6 | 74 | 29,394 | 29,496 | 3 | 13,400_13,600_100_100_29,400_29,500 | Reverse DVG | N |
| 4 | 4_7* | 13,343 | 13,425 | 6 | 74 | 29,394 | 29,496 | 3 | 13,400_13,500_100_100_29,400_29,500 | Reverse DVG | N |

investigated the clip region of the two-fragment reads, and the difference in the clip length was observed among four different protocols. In cDNA protocols, more clipped reads longer than 100 base pairs were observed, compared with the RNA protocol (Fig. S7B). To understand whether those long clip nucleotides are informative, the clipped sequences were recovered and realigned to BCoV. We found an average of 13.16% (12.32% and 14.01% from cDNA and cDNA-exo library, respectively) of the clipped sequences containing virus genome in the cDNA protocols, which was significantly higher than an average of 0.17% (0.16% and 0.18% from the RNA and RNA-exo library, respectively) in the RNA protocols (Fig. S7C). After recovering the clipped sequences, clone #1_12 could be identified from cDNA libraries and be the most abundant among all reverse DVG clone types (Fig. 5E). Based on the above findings, the bioinformatic analytical pipeline for investigating discontinuous RNA transcripts of BCoV was further amended in Fig. 4.

## DISCUSSION

Traditional approaches to studying DVGs require using Sanger sequencer to sequence the PCR product from a TA-cloned single colony, which is time-consuming and not high-throughput. ONT, as the third generation of new generation sequencer, produces long reads with high yields, which is advantageous in investigating discontinuous RNA transcripts with different recombinations or isoforms, especially RNA that can be sequenced directly (37–39). However, current studies have neither thoroughly investigated how different parameters in bioinformatic approaches affect the spectrum and accuracy in detecting coronavirus DVGs nor validated the accuracy of identified viral RNA isoforms experimentally. In this study, we chose BCoV since the experiment can be performed in a BSL-2 lab unlike SARS-CoV-2 requiring BSL-3 safety standard. We systematically evaluated different parameter settings and proposed the optimal cutoffs in identifying discontinuous RNA transcripts. Moreover, we discovered the novel reverse discontinuous RNA transcripts (reverse DVGs), which could be functional due to their translatability (40). An additional protocol to identify such reverse DVGs is recommended in the bioinformatics pipeline (Fig. 4). After comparing four different library construction protocols and specifically evaluating the cross-protocol performance in sequencing read mappability, read length, and reported base quality, no differences were observed, although the abundancy was lower when the samples were pre-treated by exonuclease in both RNA and cDNA libraries (Table S2). The lower abundancy after exonuclease treatment in both RNA and cDNA libraries could be due to the depletion of cap-free viral RNA by 5′ phosphate-dependent exonuclease (41), which leads to fewer types of DVGs observed (Fig. 2C). Moreover, the possible explanations for the highest proportion of reads mapped to BCoV using RNA without exonuclease-treatment protocol could be the biased types and less abundancy of the transcripts introduced during either the RT-PCR step (28) or additional clean-up steps during cDNA library construction.

Using minimap2 for mapping the Nanopore-generated reads is commonly used to study the CoV transcriptome landscape since it can accommodate long and short reads (42). Previous studies used arbitrary parameter settings to identify discontinuous RNA transcripts of CoV infections (29–31). In this study, we showed that the use of a bigger k-mer size tends to recover fewer recombination fragments (Fig. 2A). The main reason is that the short first fragment containing ~70-bp leader sequence for sgmRNA tends to be clipped during the alignment, which results in being classified into one fragment reads, especially when performing the direct RNA sequencing protocol (Fig. S2B). While setting k-mer at 15 is optimal to identify reproducible and reliable DVGs, a smaller k-mer size could avoid clipping the leader sequences from sgmRNA. The results were not affected by the alignment score cutoff at 60 (data not shown). Interestingly, we observed the leader sequence-containing first fragment could have up to 90 continuous nucleotides, which was longer than the expected ~70 nucleotide leader and commonly observed being fused to the N-protein open reading frame (Fig. S8A and B). By choosing 25 nucleotides as the bin size to denoise the junction regions, five types of sgmRNA

recombination to generate N proteins were observed with the dominant type being the traditionally defined coordinate, 75–249,400, followed by 100–29,400 coordinates (Fig. S8C). The discontinuous transcription into sgmRNA is mediated by the TRS containing the common 5′ "leader" sequence of ~70 nucleotides fused to the "body" sequence from the downstream part of the genome (9, 43). TRS is made up of a core sequence, as well as 5′ and 3′ flanking sequences with a leader sequence (TRS-L) and a body sequence (TRS-B) complementary to each other. It is well characterized that the dense population of RNA structures located at the 5′-terminal of the viral genome, including TRS-L, are highly conserved and home to *cis*-acting structures that regulate the activity of RNA; therefore, they are critical for viral replication (44). Many such structures have been identified as a result of the DiRNA experiment (11). How the longer TRS-L affects mRNA transcription requires further study to better understand its biological impact on the coronavirus life cycle. Our benchmark study provided a thorough evaluation of different k-mer sizes on their effect on discontinuous RNA transcripts landscape classification, which was previously unidentified using traditional sgmRNA definition.

Clip is a necessary procedure for efficient alignment by trimming the 5′ and 3′ terminals as these regions are low quality or contain the sequencing adapters (45). Although there is no adapter existing at the 5′ terminal in direct RNA library, it is known that ONT-generated reads have the highest error rate at the terminals (46). These will activate the clip procedure by the aligner because of the low mappability. We found that a higher number of reads with the longer clipped sequence were observed in the direct cDNA protocols (Fig. S7B). These clipped sequences might be another informative part of the reads. Indeed, when the clipped sequences were recovered and realigned, 13.16% of them from the cDNA libraries were mapped to the BCoV genome. After being re-assigned to BCoV, the reversed DVGs (clone #1_12-like) can be observed in the cDNA protocols (Fig. 5E). To the best of our knowledge, our study is the first study to report detecting an extra type of DVG, reversed DVGs, which is validated by both traditional TA cloning and Nanopore sequencing. Accumulated studies suggested that DVGs can (i) interfere with full-length viral genome production when accumulated to a high level, thus, resulting in viral load reduction; (ii) have strong IFN stimulation; and (iii) promote viral persistence (47). Whether the reverse DVGs identified here have similar functional interference requires further investigation.

We cannot completely rule out the possibility that the majority of reverse DVGs found from the cDNA protocols were artifacts generated due to template switching from the reverse transcription step during the library preparations (48, 49). The re-aligned clipped sequences from DVGs or sgmRNAs can not only be aligned to the BCoV genome but also to the human or RCS sequence with a higher percentage observed in the cDNA library protocol (Fig. S7D and E), suggesting the possibility of artifacts from the RT step. To further ensure that this observation can be found in other coronaviruses, we downloaded the published data set (https://doi.org/10.17605/OSF.IO/8F6N9), which analyzed the SARS-CoV-2 RNA transcripts by Nanopore sequencer using the same DRS protocol (30). After processing the reads using the same pipeline developed here, RCS can also be identified from the clipped nucleotide sequences. This observation emphasizes the potential of generating artifacts using direct RNA or direct cDNA protocols. Currently, very few studies have explored the reversed and chimeric transcripts in the CoV life cycle (50, 51). Future study is required to understand if the reverse DVGs are transcribed due to unexpected RNA secondary structure formation.

Previous studies investigated the landscape of RNA transcriptomes by ONT (29–31); however, this is the first study that has experimentally validated long-read sequencing results. Other than sgmRNA, the rest of the truncated form of the viral genome, including DiRNA, will be grouped as DVGs since we do not perform functional assays for DVGs. Out of 10 colonies selected, our colony PCR sequencing results only identified one type of sgmRNA, which is N-protein, three types of DVGs, and five types of reverse DVGs (Table 2). Furthermore, based on the clonally identified discontinuous transcript types, we could firmly compare the reproducibility and abundance among different protocols.

The identified DVG clone #1_2 could be found among four protocols with a higher abundance from the direct RNA sequencing protocol. Furthermore, the results showed that the treatment of exonuclease reduced the abundance for both RNA and cDNA protocols.

Although most people infected with SARS-CoV-2 develop a mild to moderate disease with virus replication restricted mainly to the upper airways, some progress to having a life-threatening acute respiratory distress syndrome with predispositions leading to immunopathology (52). *In vitro* and clinical studies have shown that SARS-CoV-2 is a poor inducer of interferons (53). Patients with mild COVID-19 disease showed extensive induction of type I and III interferon responses, whereas patients with severe disease demonstrated poor response in their antiviral capacity despite higher local inflammatory myeloid cell populations and equivalent viral loads (54, 55). Higher local inflammatory response of the epithelium and endothelium triggers the imbalance between the activation of coagulation and the inhibition of fibrinolysis (56). Such a cascade of inflammation and coagulation interplayed among monocytes, macrophages, and neutrophils was further amplified and eventually led to severe immunopathology (57). Corticosteroids are frequently used as general inhibitors of inflammation, and a 50% reduction in mortality by administration of dexamethasone to patients with severe COVID-19 was observed (58). A neutralizing monoclonal antibody to IL-6 was also shown to increase survival in hospitalized patients (59). On the contrary, direct antiviral strategies are effective only when they are administered very early after infection. Remdesivir, molnupiravir, and bemnifosbuvir are the nucleic acid analogs designed to combat SARS-CoV-2 by terminating the viral RNA synthesis (60–62). However, while they are effective in preventing hospitalization and death in outpatients, they are ineffective in patients already hospitalized with COVID-19 (63). Naturally arising viral particles with defective viral genomes have been characterized in several coronavirus members and are responsible for the emergence of the new viral variants (40, 64). Importantly, studies have shown such DVGs, naturally occurring or synthetic, are capable of reducing viral RNA levels by competing for cellular or viral resources and could serve as anti-viral agents (65, 66). The algorithms developed in this study could serve as the platform for identifying functional and interfering DVGs.

In this benchmarking study, we compare the k-mer size, gap size, segment size, and bin size to give the overall picture of how the parameters affect the discontinuous transcriptome in four protocols. Four novel findings are summarized, including (i) optimal bioinformatic pipeline for investigating discontinuous RNA transcripts proposed in Fig. 4 with both sensitivity and specificity reaching 82.6% and 99.6%, respectively, under k-mer size at 15; (ii) no exonuclease treatment is needed for library preparation; (iii) both RNA and cDNA direct sequencing gave similar DVG diversity results although the abundancy is slightly higher from the RNA direct sequencing protocol; and (iv) additional recovery of clipped nucleotide sequences could expand the landscape of CoV discontinuous RNA transcriptome.

## MATERIALS AND METHODS

### Cells and viral infection

BCoV strain BetaCoV/Korea/KCDC03/2020 (GenBank accession no. U00735.2), which was obtained from David A. Brian (University of Tennessee, TN, USA), was plaque-purified and grown in human rectum tumor (HRT)-18 cells. HRT-18 cells (ATCC CCL-244) were maintained in Dulbecco's modified Eagle's medium supplemented with 10% fetal bovine serum (Hyclone, UT, USA) at 37°C with 5% $CO_2$.

### RNA sample extraction and enrichment

HRT-18G cells were infected with the BCoV strain at a multiplicity of infection of 0.1. After 48 hours, the infected cells were harvested by removing the culture supernatant,

followed by PBS wash for one time, and treated with trypsin for 3 min. The trypsinized cells were then centrifuged at 10,000 × *g* for 15 min using low-speed centrifugation. Then, the obtained cells were lysed by the TRIzol reagent (TR 118, MRC), and total RNA was extracted from the aqueous phase. The extracted total RNA was enriched by oligo-d(T)25 magnetic beads (S1408, New England BioLabs) for the poly-A tail containing transcripts. After the poly-A enrichment, the total RNA was quantified by the Qubit with RNA high sensitivity kit (Q32852, Thermo Fisher) and stored at −80°C for future use.

## RNA sample pre-processing and Nanopore library construction

Figure 1 illustrates four different protocols applied to generate transcriptome data with two replicates for each protocol, which generated eight different libraries in total. The total poly-A-enriched RNAs from the same infected cell lysate were used for library constructions. To test if RNA with incomplete 5′ UTR would affect the RNA and cDNA library construction, 5′-phosphate-dependent terminator exonuclease was used to remove the ribosomal RNA and non-capped degraded messenger RNA based on the manufacturer's protocol (TER51020, Lucigen). In brief, 17 µL of RNA, 0.5 µL of 5′-phosphate-dependent exonuclease, and 2 µL of 10× buffer A were mixed gently in a 0.2 µL PCR tube. The mixture was incubated at 30°C for 60 min and then cleaned up by 1× kapa pure beads (7983271001, Roche) to remove excess enzyme and small fragment RNA.

For direct RNA sequencing, 500 ng of RNA was carried into the library preparation using the Oxford Nanopore DRS protocol (SQK-RNA002, Oxford Nanopore). The unmethylated RNA calibration standard, derived from the yeast enolase II (YHR174W) gene, was provided by the kit and added as a negative control during the standard RNA library preparation. For Nanopore cDNA sequencing, 250 ng of RNA was applied for library preparation using the Oxford Nanopore cDNA library protocol (SQK-DCS109, Oxford Nanopore). All steps were followed according to the manufacturer's specifications. The library was then loaded on an R9.4.1 flow cell and sequenced on a MinION device (ONT). The sequencing run was terminated after 48 hours.

## Nanopore data processing and species mapping

The fast5 files were basecalled and transformed to fastq files by guppy [v6.0.0 (67)] with custom settings. The minimum score required for read quality was set to 7. The initial intention includes comparing different aligners, such as Burrows-Wheeler Aligner (68), bbmap (69), Spliced Transcripts Alignment to a Reference (70), GraphMap (71), or minimap2, and the initial results showed that only minimap2 yielded the highest reads mapped to BCoV. Therefore, minimap2 was used for the analysis here. Raw reads coverage and sequence identity were determined from alignment to the reference sequences, the human genome (Genecode, GRCh38), bovine coronavirus reference (GenBank, U00735.2), and yeast enolase II RCS (GenBank, NP_012044.1) by minimap2 (v2.24) (42) with loose parameters of "-k 14 -w 1 -u n --MD -a -t 24 --secondary=no" and without splicing mode setup for comprehensively mapping every read from each library. CIGAR strings were used to identify the insertion/deletion compared to the BCoV reference sequence, and error rates were further computed using ALFRED (72).

## Characterization of discontinuous transcript variants and sgmRNAs of BCoV

To fully uncover the reads mapped to BCoV genome with different splicing forms, the basecalled fastq-transformed reads were aligned to BCoV reference sequence by minimap2 with splicing parameters of "-k 15 -w 1 --splice -g 30000 -G 30000 -F 40000 -N 32 --splice-flank=no --max-chain-skip=40 -u n --MD -a -t 24 --secondary=no" to comprehensively investigate the genomic RNA landscape of BCoV. Different k-mers from "-k 8" to "k 27" were used to decide the optimal cutoff. The output files were polished by the TranscriptClean [v2.0.3 (73)] to remove the mismatches and microindels. The aligned sam files were transformed to bed format by "samtools" [v1.16.1 (74)] and "bedtool"

[v2.27.1 (75)] with the "split" parameter. The BED file parsing the BAM BCoV alignments into viral genomic positions determines the start, end, and junction coordinates of the alignments. The analysis was performed based on these read coordinates.

Sequencing read depth was calculated by "samtools" depth. The reads classification was processed by following steps, reads were reflected in their viral sequence using "bedtool" and "biotools-master" by bed file and reference. These reflected sequences are utilized to predict the ORF and the translated peptides. A start codon located at the coordinate 100 of the BCoV viral genome is non-translatable due to the secondary structure at the 5′ non-coding region. If the reads contain this region, the second start codon will be used for translation into peptides or proteins. The derived peptides were used to accurately classify the reads into the known can_sgmRNA, translatable defective viral genome, or untranslatable DVGs. Those with 100% identity to known structural proteins were classified as sgmRNA, and the identified sgmRNA should contain the 5′ 70 nucleotide-long "leader" sequence. The rest of the truncated forms of the viral genome were grouped as DVGs. The two-fragment reads are recorded as viral genomic sites in a vector such as (read_start, break1_5′, break1_3′, read_end) for bin size analysis.

## Parameter comparison

Four different parameters including k-mer size, gap size, segment size, and bin size were compared in this study. The k-mer size is the parameter used in the minimap2 aligner. During alignment, the reference is cleaved into oligonucleotides using "k-mer length" size. When the k-mer size is 8, the reference is cleaved into eight oligonucleotide length. The discontinuous transcripts contain two features, which are gap size and fragment size. The gap size denotes the length between two discontinuous fragments, while the fragment size is the length of fragments of the mapped BCoV RNA transcripts. Bin size is the interval used to group the types of recombination based on the coordinates of the BCoV viral genome to avoid the randomness of sequencing and alignment.

## RT-PCR and TA cloning

The total RNA extracted from cell lysate as described in the "RNA sample extraction and enrichment" section was reverse transcribed to complement DNA by random hexamer using the SuperScript III First-strand synthesis kit (Invitrogen, CA, USA). We focused on four types of N protein-containing DVGs with different splicing fragments, which can be identified by all four different libraries with high abundance for primer design and PCR amplification based on our previous experiences (11). The primer sequences are provided in Table S1. The estimated sizes of PCR products are Type I: 395 bp, Type II: 650 bp, Type III: 400 bp, and Type IV: 450 bp. The predicted length was used as guidance to perform gel cutting and purification. In the first round of PCR, 2 µL of cDNA, 2 µL of primer pairs (10 µM), dNTPs (2.5 mM), and ProTaq DNA polymerase (Protech) were mixed well and ran through the thermocycling conditions consisting of 2 min at 94℃, 35 cycles of 30 seconds each at 94℃, 55℃, 72℃, and 5 min at 72℃. The PCR products amplified by four different primer pairs were confirmed by electrophoresis using 1% DNA agarose gel. The DNA products with the expected sizes were cut and purified by NucleoSpin Gel and PCR clean-up kit (MACHEREY-NAGEL) according to the manufacturer's guidance. The final products were subjected to the second-round PCR using the same primer pairs under the same PCR conditions followed by gel extraction of the expected DNA sizes. The purified products were cloned into the TOPO vector (Invitrogen) and transformed into *Escherichia coli 10 G* competent cells (Biosearch Technologies) by electroporation. The conditions consisted of voltage: 1,250 V, capacity: 25 µF, and resistance 200 Ohm. Four hundred microliters of culture medium was added for recovery and shaken at 37℃ at 225 rpm for 1 hour. After 1 hour, 200 µL of transformed competent cells were plated on a 2× YT plate containing 50 µg/mL of kanamycin and incubated at 37℃ overnight. The next day, 24 colonies were picked randomly, and colony PCR was performed using the first-round PCR primers to confirm the insert size. Three, three, five, and four colonies containing the expected insert sizes, as predicted from Types I-IV, respectively, were subjected to

traditional Sanger sequencing. The sequences obtained from Sanger sequencing were further aligned to the BCoV viral genome using the blast tool from NCBI.

## Simulation analysis

A total of 20,000 Nanopore-simulated reads for DVGs were generated with expected error profiles by randomly introducing 15% errors, including point mutation, deletion, and insertion using R script as shown in Fig. S6B. The reads were generated by using the following conditions: (i) the first fragment with a length ranging from 3 to 1,000 bp is located at nucleotide position 1–1,000 of the 5′ terminal of BCoV viral genome; (ii) the second fragment is located at the nucleotide position 28,000–31,032 with a length ranging from 3 to 3,000; (iii) the simulated DVG reads are translatable but the proteins after translation are not one of the full-length BCoV viral proteins. After running the R scripts, 500 different types of DVGs were generated for each library. Next, we generated 100 DVG libraries using the bootstrapping approach by performing resampling with replacement with the same library size of 20,000 reads and repeating 100 times. Then, we tested the sensitivity of classifying DVGs from the simulated reads using the proposed pipeline in Fig. 4 by varying k-mer sizes from 8 to 27. In order to calculate specificity, we generated another 20,000 Nanopore-simulated reads for sgmRNA with the same 15% expected error profiles using the R script in Fig. S6B. The sgmRNA reads by definition were generated by using the following conditions: (i) the first fragment must include a leader sequence with a length ranging from 3 to 70 nucleotides; (ii) the second fragment would be one of the known ORFs at the 3′ end of the genome, which could be translated to structural or accessory proteins of BCoV, including ORF1ab, 32k, HE, S, 4.9k, 4.8k, 12.7k, E, M, and N with 10 different sgmRNA types in total. The 100 bootstrapped sgmRNA libraries were used as negative control and ran through the same proposed pipeline for specificity.

## Statistical analysis

Analysis of variance was calculated to examine the statistical difference of the aligned reads to different species among different protocols and the Tukey's honest significance test was performed to examine the significance between groups by using the R package (v4.2.2) (76). Figures were generated with ggplot2 (77) and ComplexHeatmap (78). DVG maps were generated using the gggenes (79) and ggtranscripts (80) in the R package.

### ACKNOWLEDGMENTS

This work was supported by grants (MOST 110-2327-B-005-524 003 and 110-2811-B-005-002) from the Ministry of Science and Technology (MOST), R.O.C.

We thank Dr. David A. Brian at the University of Tennessee, Knoxville, USA for providing BCoV and MHV-A59.

### AUTHOR AFFILIATIONS

[1]Doctoral Program in Microbial Genomics, National Chung Hsing University and Academia Sinica, Taichung, Taiwan

[2]Graduate Institute of Veterinary Pathobiology, College of Veterinary Medicine, National Chung Hsing University, Taichung, Taiwan

[3]Department of Veterinary Integrative Biosciences, Texas A&M University, College Station, Texas, USA

[4]Graduate Institute of Microbiology and Public Health, College of Veterinary Medicine, National Chung Hsing University, Taichung, Taiwan

[5]Department of Post-Baccalaureate Medicine, College of Medicine, National Chung Hsing University, Taichung, Taiwan

## AUTHOR ORCIDs

Bo-Jia Chen [iD] http://orcid.org/0000-0002-0439-8488
Hung-Yi Wu [iD] http://orcid.org/0000-0002-1260-6259
Day-Yu Chao [iD] http://orcid.org/0000-0001-7139-026X

## FUNDING

| Funder | Grant(s) | Author(s) |
| --- | --- | --- |
| National Science and Technology Council (NSTC) | MOST 110-2327-B-005 -524 003 | Bo-Jia Chen |
| National Science and Technology Council (NSTC) | 110-2811-B-005 -002 | Ching-Hung Lin |

## AUTHOR CONTRIBUTIONS

Bo-Jia Chen, Data curation, Formal analysis, Software | Ching-Hung Lin, Data curation, Formal analysis, Investigation, Methodology | James J. Cai, Conceptualization, Supervision, Writing – review and editing | Day-Yu Chao, Conceptualization, Funding acquisition, Investigation, Methodology, Project administration, Resources, Supervision, Validation, Writing – original draft, Writing – review and editing.

## DATA AVAILABILITY

This study's basecalled long-read RNA (ONT) data have been submitted to the NCBI SRA under accession number PRJNA941120. All analysis codes are available on GitHub (https://github.com/BJ-Chen-Eric/BCoV-SV).

## ADDITIONAL FILES

The following material is available online.

### Supplemental Material

**Fig. S1-S8, Tables S1 and S2 (Spectrum03954-23-s0001.pdf).** Supplemental figures and tables.

### Open Peer Review

**PEER REVIEW HISTORY (review-history.pdf).** An accounting of the reviewer comments and feedback.

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
