## [Reviewer comments · Microbiology Spectrum]

Microbiology Spectrum

Experimental and analytical pipeline for sub-genomic RNA landscape of coronavirus by Nanopore sequencer

BoJia Chen, Ching-Hung Lin, Hung-Yi Wu, James Cai, and Day-Yu Chao

Corresponding Author(s): Day-Yu Chao, National Chung Hsing University

Review Timeline:

Submission Date:	November 15, 2023
Editorial Decision:	January 30, 2024
Revision Received:	February 22, 2024
Accepted:	February 26, 2024

Editor: Biao He

Reviewer(s): Disclosure of reviewer identity is with reference to reviewer comments included in decision letter(s). The following individuals involved in review of your submission have agreed to reveal their identity: Pawan Kumar RAGHAV (Reviewer #1)

Transaction Report:

DOI: <https://doi.org/10.1128/spectrum.03954-23>

Re: Spectrum03954-23 (Experimental and analytical pipeline for sub-genomic RNA landscape of coronavirus by Nanopore sequencer)

Dear Dr. Day-Yu Chao:

Thank you for the privilege of reviewing your work. Below you will find my comments, instructions from the Spectrum editorial office, and the reviewer comments.

This is an interesting study providing a methodological exploration of the viral RNA transcriptome. Appended are some minor pieces of advice for your consideration. Happy Chinese New Year!

Revision Guidelines

Sincerely,
Biao He
Editor
Microbiology Spectrum

Reviewer #1 (Comments for the Author):

The manuscript "Experimental and analytical pipeline for sub-genomic RNA landscape of coronavirus by Nanopore sequencer" reported the study on bovine coronavirus (BCoV) utilized four different ONT protocols and rigorous bioinformatic analysis, determining optimal parameters for identifying defective viral genome (DVG) reads. The proposed pipeline demonstrated high

sensitivity and specificity in detecting informative DVG reads, offering benchmarks for studying discontinuous coronavirus RNA transcriptomes. The manuscript is well written and complete in all aspect. Here are my minor suggestions given below to improve the quality of current version of manuscript:

1. The mechanism of pathogenesis of SARS-CoV-2 viral genome is not discussed in the study as discussed in the following study. <https://doi.org/10.1016/j.mehy.2020.110031>

2. The structural information of drug targets, RBD and receptor domains; ACE2 mutations which are resistant to COVID-19 infection should also be discussed are missing which must be discussed in the introduction as discussed in the following studies. <https://doi.org/10.1038/s41598-022-20773-9>
<https://doi.org/10.1007/s00726-021-02991-z>

3. There are molecules used to combat COVID-19, which should also be discussed in the current study that is related to the study.

<https://doi.org/10.1016/j.jphs.2023.02.004>

<https://doi.org/10.1016/j.mehy.2020.110031>

Reviewer #3 (Comments for the Author):

1. It is recommended to briefly introduce the role of 5' phosphate-dependent exonuclease processing in the Introduction, along with an explanation of using the RCS, which may give readers a clearer understanding of the experimental design.
2. As mentioned in the Results, "Higher reads from RNA libraries were observed than those from cDNA libraries." It may be due to the higher initial input of DRS than others. Using the same initial input for this comparison will be more convincing.
3. "In particular, protocol #2 (RNA without exonuclease-treatment protocol) showed the highest proportion of reads mapped to BCoV (Supplementary Fig S1C)." A further discussion to explain the reason is recommended.
4. In comparing RNA_exo1 and RNA1, the number of Human host reads increased by the exonuclease processing, while the opposite result was found in comparing RNA_exo2 with RNA2. Is it reasonable to explain this only in terms of "the removal of highly abundant rRNA, which leaves more pores free for RNA sequencing"?
5. In Figure 2, legend titles should be added to the graph, such as "number of fragment reads" and "k-mer size," the same applies to Figure 3.
6. Keep the aspect ratio when stretching the image in Figure 5.
7. The image resolution in the Supplementary needs to be improved.
8. Several grammatical mistakes should be corrected.

Dear Editor,

We would like to thank the reviewers for their constructive comments. We have prepared a revised version of our manuscript and answered the questions/comments raised by the reviewers point-by-point below. Revisions in the manuscript were also highlighted in red.

Comments from Reviewer #1

1. The mechanism of pathogenesis of SARS-CoV-2 viral genome is not discussed in the study as discussed in the following study.

Reply:

We are thankful for the reviewer's comment and such discussion is appended in page 17, line 467-479 and page 5-6, line 102-111 as the following.

- Although most people infected with SARS-CoV-2 develop a mild to moderate disease with virus replication restricted mainly to the upper airways, some progress to having a life-threatening acute respiratory distress syndrome (ARDS) with predispositions leading to immunopathology [54]. In vitro and clinical studies have shown that SARS-CoV-2 is a poor inducer of interferon [55]. Patients with mild COVID-19 disease showed extensive induction of type I and III interferon responses, whereas patients with severe disease demonstrated poor response in their antiviral capacity despite higher local inflammatory myeloid cell populations and equivalent viral loads [56, 57]. Higher local inflammatory response of the epithelium and endothelium triggers the imbalance between the activation of coagulation and the inhibition of fibrinolysis [58]. Such cascade of inflammation and coagulation interplayed among monocytes, macrophage and neutrophils were further amplified and eventually lead to severe immunopathology [59]. Corticosteroids are frequently used as general inhibitors of inflammation and 50% reduction in mortality by administration of dexamethasone to patients with severe COVID-19 was observed [60]. A neutralizing monoclonal antibody to IL-6 were also shown to increase survival in hospitalized patients [61].
- There are kinds of cis-acting elements, required for viral genome transcription, gene expression and pathogenesis, in the 5' and 3' termini of coronaviral genome. The cis-acting elements in the 5' terminal of the genome are composed of multiple stem-loops (SLs), including SL I to VII. While the cis-acting elements in the 3' end of the genome consist of a

bulged stem loop (BSL), pseudoknot (PK), hypervariable region (HVR) and poly(A) tail [14, 15]. Functional DVGs (so-called DiRNA), if contain the cis-acting elements required for translation or replication, may play different roles in triggering cell innate immune signaling, interfering with the sgRNA transcription/translation, or assisting in establishing persistence infection [16, 17].

2. The structural information of drug targets, RBD and receptor domains; ACE2 mutations which are resistant to COVID-19 infection should also be discussed are missing which must be discussed in the introduction as discussed in the following studies.

Reply:

We are thankful for the reviewer's comment and further introduction is appended in page 6, Line 113-129 as the following.

- The S protein which mediates the binding with cellular receptors for infection are a characteristic feature of the *coronaviridae* family. Both SARS-CoV-2 and SARS-CoV bind to a common human receptor, angiotensin-converting enzyme 2 (ACE2), which is also the receptor for other human CoVs except MERS-CoV [18]. The S protein consists of two subunits: The S1 unit at the N-terminus of S protein forms the head that contains receptor-binding domain (RBD) is responsible for cellular receptor binding; whereas the S2 unit presents in the stalk of S protein mediating the fusion process for viral entry [18]. These two subunits are separated by the site, which contains a furin cleavage motif and is cleaved by the transmembrane serine protease TMPRSS2 in the virus-producing cell [19]. This cleavage activates the S2 subunit trimers to fuse viral and host lipid bilayers, releasing the viral ribonucleoprotein complex into the cell. Amino acid variations in human ACE2 proteins have been suggested to mediate RBD binding affinity, which could either enhance or inhibit virus entry [20]. As such, vesicles designed to carry the S protein or RBD could be used to antagonize virus entry [21-23]. Alternatively, extracellular vesicles (EVs), derived from stem cells that carry ACE2, could be used to treat infections by coronaviruses [24, 25].

3. There are molecules used to combat COVID-19, which should also be discussed in the current study that is related to the study.

Reply:

We are thankful for the reviewer's comment and further discussed in page 17-18, Line 479-494 as the following.

- Corticosteroids are frequently used as general inhibitors of inflammation and 50% reduction in mortality by administration of dexamethasone to patients with severe COVID-19 was observed [60]. A neutralizing monoclonal antibody to IL-6 were also shown to increase survival in hospitalized patients [61]. On the contrary, direct antiviral strategies are effective only when they are administered very early after infection. Remdesivir, Molnupiravir and Bemnifosbuvir are the nucleic acid analogs designed to combat the SARS-CoV-2 by terminating the viral RNA synthesis [62-64]. However, they are effective against hospitalization and death in outpatients but ineffective in patients hospitalized with COVID-19 [65]. Naturally arising viral particles with defective viral genome (DVGs) have been characterized in several coronavirus members and are responsible for the emergence of the new viral variants [42, 66]. Importantly, studies have shown such DVGs, naturally occurred or synthetic, are capable of reducing viral RNA levels by competing the cellular or viral resources and could serve as the anti-viral agents [67, 68]. The algorithms developed in this study could serve the platform identifying functional and interfering DVGs.

Comments from Reviewer #3

1. It is recommended to briefly introduce the role of 5' phosphate-dependent exonuclease processing in the Introduction, along with an explanation of using the RCS, which may give readers a clearer understanding of the experimental design.

Reply:

We are appreciated with the reviewer's comment and the revision are made for clarity in Page 7, line 153-158 and Page 8, Line 181-182 as the following.

- Since the majority of host messenger RNA (mRNA) or viral RNA transcripts are protected from degradation by m7Gppp cap and triphosphate [36], the 5' phosphate-dependent exonuclease, by removing the RNA population with 5' monophosphate group such as ribosomal RNA (rRNA) and transfer RNA (tRNA), was tested in this study for its influence on BCoV transcriptome.
- As the unmethylated RNA calibration standard (RCS), used to assess the false detection rate of the methylation calling of RNA molecule, is only offered in DRS kit, 53.3 and 26.7% of reads on average from protocol #1 and #2 were mapped to RCS with higher RCS reads proportion observed in

exonuclease treatment libraries (Table 1).

2. As mentioned in the Results, "Higher reads from RNA libraries were observed than those from cDNA libraries." It may be due to the higher initial input of DRS than others. Using the same initial input for this comparison will be more convincing.

Reply:

As reviewer mentioned, using the same initial input for this comparison will be more convincing. However, Nanopore ONT Minion requires strict input amounts to the chips, which is 500 ng in direct RNA sequencing (DRS) and 250 ng in direct cDNA sequencing (DCS). Higher or lower input may affect the sequencing performance. Therefore, each protocol has their own initial input suggested by the manufacturer to help achieve the final product amount loaded to the chips in this study. Furthermore, we didn't observe any difference in the percentages of fragment numbers or DVG types between DRS and DCS under the pre-set cutoff of different parameters. We are appreciated with the reviewer's comment and clarification is made in Page 8, line 176-177 as the following.

- Higher reads from RNA libraries were observed than those from cDNA libraries, possibly due to higher initial RNA input during the library constructions as suggested by the manufacture's protocols.

3. "In particular, protocol #2 (RNA without exonuclease-treatment protocol) showed the highest proportion of reads mapped to BCoV (Supplementary Fig S1C)." A further discussion to explain the reason is recommended.

Reply:

We are thankful for the reviewer's comment and further discussion is appended in page 14, line 377-383 as the following.

- The lower abundancy after exonuclease treatment in both RNA and cDNA libraries could be due to the depletion of cap-free viral RNA by 5' phosphate-dependent exonuclease [43], which leads to fewer types of DVG observed (Fig 2C). Moreover, the possible explanations of highest proportion of reads mapped to BCoV using RNA without exonuclease-treatment protocol could be the biased types and less abundancy of the transcripts introduced during either the RT-PCR step [29] or additional clean-up steps during cDNA library construction.

4. In comparing RNA_exo1 and RNA1, the number of Human host reads increased by the exonuclease processing, while the opposite result was found in comparing RNA_exo2 with RNA2. Is it reasonable to explain this only in terms of "the removal of highly abundant rRNA, which leaves more pores free for RNA sequencing"?

Reply:

By reviewing the literature, 20-30% of RCS among the total reads is very commonly observed. On the contrary, the high amount of RCS (46-76%) in RNA-exo group found in this study is not common. The possible explanation of the effect of 5' phosphate-dependent exonuclease could be the removal of the abundant rRNA, which leaves more pores free for RNA sequencing as original stated in the manuscript. Another explanation could be the RNA degradation during the process. We are appreciated with the reviewer's comment and minor revision are made for clarity in page 8, line 187-189 as the following.

- Therefore, the relatively high percentage of RCS found in protocol #1 could be due to the removal of highly abundant rRNA, which leaves more pores free for RNA sequencing, especially RCS, although the possibility of a higher percentage of RNA degradation by exonuclease treatment could not be ruled out.

5. In Figure 2, legend titles should be added to the graph, such as "number of fragments reads" and "k-mer size," the same applies to Figure 3.

Reply:

We are thankful for the reviewer's advice. The legend titles are appended to Figure 2 and Figure 3 as suggested

6. Keep the aspect ratio when stretching the image in Figure 5.

Reply:

Thanks for the reviewer's comment. We re-produce Figure 5 without stretching with the correct aspect ratio.

7. The image resolution in the Supplementary needs to be improved.

Reply:

Thanks for the reviewer's kind advice. We reproduce all the supplementary

figures and are sure they meet the journal criteria.

8. Several grammatical mistakes should be corrected.

Reply:

We appreciated the reviewer's comment. The manuscript was edited by the native English speaker and the grammatical mistakes have been properly corrected.

Re: Spectrum03954-23R1 (Experimental and analytical pipeline for sub-genomic RNA landscape of coronavirus by Nanopore sequencer)

Dear Dr. Day-Yu Chao:

Your manuscript has been accepted, and I am forwarding it to the ASM production staff for publication. Your paper will first be checked to make sure all elements meet the technical requirements. ASM staff will contact you if anything needs to be revised before copyediting and production can begin. Otherwise, you will be notified when your proofs are ready to be viewed.

Sincerely,
Biao He
Editor
Microbiology Spectrum